# Investigation of Mutations in the *crt-o* and *mdr1* Genes of *Plasmodium vivax* for the Molecular Surveillance of Chloroquine Resistance in Parasites from Gold Mining Areas in Roraima, Brazil

**DOI:** 10.3390/microorganisms12081680

**Published:** 2024-08-15

**Authors:** Jacqueline de Aguiar Barros, Fabiana Granja, Rebecca de Abreu-Fernandes, Lucas Tavares de Queiroz, Daniel da Silva e Silva, Arthur Camurça Citó, Natália Ketrin Almeida-de-Oliveira Mocelin, Cláudio Tadeu Daniel-Ribeiro, Maria de Fátima Ferreira-da-Cruz

**Affiliations:** 1Malaria Control Center, Epidemiological Surveillance Department, General Health Surveillance Coordination, SESAU-RR, Boa Vista 69310-043, RR, Brazil; barros.jacqueline@gmail.com; 2Center for Biodiversity Studies, Federal University of Roraima (UFRR), Boa Vista 69310-000, RR, Brazildanieldas.silva@yahoo.com (D.d.S.e.S.); 3Graduate Program in Biodiversity and Biotechnology (Bionorte-RR), Boa Vista 69301-290, RR, Brazil; 4Laboratório de Pesquisa em Malária, Instituto Oswaldo Cruz Fundação Oswaldo Cruz (Fiocruz), Rio de Janeiro 21040-900, RJ, Brazil; rebeccaabreuf@hotmail.com (R.d.A.-F.); lucasqueiroz@aluno.fiocruz.br (L.T.d.Q.); nataliaketrin@gmail.com (N.K.A.-d.-O.M.);; 5Center for Malaria Research, Diagnosis and Training (CPD-Mal)/Reference Center for Malaria in the Extra-Amazon Region of the Brazilian Ministry of Health, Fiocruz, Rio de Janeiro 21040-900, RJ, Brazil; 6Research Support Center in Roraima (NAPRR), National Institute for Amazonian Research (INPA), Boa Vista 69301-150, RR, Brazil; arthur.cito@inpa.gov.br

**Keywords:** Amazon, malaria, chloroquine, *pvcrt-o*, *pvmdr1*, mining

## Abstract

*Plasmodium vivax* causes the largest malaria burden in Brazil, and chloroquine resistance poses a challenge to eliminating malaria by 2035. Illegal mining in the Roraima Yanomami Indigenous territory can lead to the introduction of resistant parasites. This study aimed to investigate mutations in the *pvcrt-o* and *pvmdr-1* genes to determine their potential as predictors of *P. vivax* chloroquine-resistant phenotypes. Samples were collected in two health centers of Boa Vista. A questionnaire was completed, and blood was drawn from each patient. Then, DNA extraction, PCR, amplicon purification, and DNA sequencing were performed. After alignment with the Sal-1, the amplified fragment was analyzed. Patients infected with the mutant parasites were queried in the Surveillance Information System. Among the patients, 98% (157/164) of participants were from illegal mining areas. The *pvcrt-o* was sequenced in 151 samples, and the K10 insertion was identified in 13% of them. The *pvmdr1* was sequenced in 80 samples, and the **M**YF haplotype (958**M**) was detected in 92% of them and the TYF was detected in 8%, while the **M**Y**L** was absent. No cases of recrudescence, hospitalization, or death were found. Mutations in the *pvcrt-o* and *pvmdr-1* genes have no potential to predict chloroquine resistance in *P. vivax*.

## 1. Introduction

*Plasmodium vivax* is the second most important malaria pathogen in the world, being responsible for 2.8% of the 249 million malaria cases in 2022. Outside of Africa, it is the main *Plasmodium* species, and it has the widest geographical distribution in the world. It causes significant morbidity in Southeast Asia, North and South America, and the Middle East [1].

In the Americas, 72% of malaria cases are caused by *P. vivax*, and four countries accounted for 80% of estimated cases in 2022: the Bolivarian Republic of Venezuela (28%), Brazil (27%), Colombia (18%), and Peru (6%) [1].

In 2023, 142,522 malaria cases were reported in Brazil, of which 99.97% were autochthonous in the states of the legal Amazon region (Acre, Amapá, Amazonas, Maranhão, Mato Grosso, Pará, Rondônia, Roraima, and Tocantins), and 82.6% were caused by *P. vivax*. In the same year, 34,555 cases were reported in Roraima, 71% of which were caused by *P. vivax* [2].

To achieve the goal of malaria elimination by 2035, the National Malaria Program (PNCM) envisages only *P. vivax* transmission for the last five years after the elimination of malaria transmission by *P. falciparum* in 2030 [3]. However, the unique biology of *P. vivax* poses additional challenges to the elimination target, such as relapses caused by hypnozoites, the ability to infect mosquitoes before symptoms appear, and asymptomatic infections in endemic areas [4].

Adding to this context, the emergence of resistance to chloroquine (CQR), the drug of first choice for the elimination of the blood stages of *P. vivax*, may represent a further obstacle to control strategies based on the use of this antimalarial drug [5,6]. Indeed, three decades after the emergence of *P. falciparum* CQR, such resistance in *P. vivax* was first reported in Papua New Guinea in 1980, and subsequent studies have shown an increase in this resistance, accompanied by reports of severe and fatal *P. vivax* malaria in these regions [7,8].

In addition to Southeast Asia, particularly in Thailand [9] and Cambodia [10], *P. vivax* CQR has been documented in East Africa, Ethiopia [11,12], South America, parts of the Guiana Shield, and the Cooperative Republic of Guyana [13].

In Brazil, CQR was first detected in 1999 in a patient in Manaus in Amazonas state [14]. Later studies in the same city reported a 10.1% failure rate for treatment with CQ in 2007 [15] and 5.2% in 2014 [16]. More recently, 1.1% treatment failure following supervised treatment with CQ and primaquine (PQ) was reported in the city of Oiapoque in the state of Amapá, on the border with French Guiana [17].

The resistance mechanisms in *P. vivax* are still not fully understood, probably also because this parasite species lacks continuous in vitro cultivation methods [18,19]. In this context, monitoring with molecular markers has emerged as a practical and cost-effective field tool for antimalarial drug resistance monitoring compared to in vivo and in vitro tests [18].

Molecular markers that can be considered for predicting a malaria resistance phenotype in *P. vivax* have been identified based on the orthologs’ resistant-related genes in *P. falciparum* [20,21]. Thus, the multidrug resistance gene 1 of *P. vivax* (*pvmdr1*) and the CQ resistance transporter gene (*pvcrt-o*) are orthologous to the *pfmdr1* and *pfcrt* genes in *P. falciparum* [18,22,23]. Alterations in the *pvmdr1* sequences are thought to confer CQR by reducing the transport of CQ into the digestive vacuole (DV), where the parasite digests host cell proteins and converts hemoglobin heme to nontoxic hemozoin [24].

The mutant *pvcrt*, in turn, would act as an efflux pump in the active transport of CQ out of the DV and away from its target (converting heme to hemozoin) [18]. The lysine insertion (AAG) in the first exon (amino acid 10), referred to as the K10 insertion in the *pvcrt-o* gene, would be associated with a reduction of half of the maximum inhibitory concentration of CQ (IC50) and has been identified as a possible molecular marker for CQR in *P. vivax* [25,26]. Concerning *pvmdr1*, the amino acid mutations Y976F, F1076L, and T958M have been linked to CQR [21,26,27].

In Roraima, the malaria burden in the state has increased, especially since 2018, due to the increased migration flow from Venezuela and Guyana, together with the boost of illegal mining in the Yanomami Indigenous land. In the same period, there has been an increase in hospitalizations and deaths by *P. vivax* malaria [28], and it is known that the clinical severity of malaria could be related to the appearance of CQR [29].

Since little is known about the *P. vivax* genotypes of *pvmdr1* and *pvcrt-o* genes circulating in the parasites of Brazilian endemic areas, such as those of Roraima, this study aimed to investigate the polymorphism of these genes in regions with a great influx of people.

## 2. Materials and Methods

This study was approved by the Research Ethics Committee of the Federal University of Roraima CAAE 24122619.6.0000.5302 (CEP/UFRR, acronym in Portuguese; opinion n. 3,920,373, issued on 17 March 2020). The CEP/UFRR allowed only the inclusion of non-village Indigenous people who speak Brazilian Portuguese and reside in Boa Vista. Additionally, the research project was demanded to be presented to the Kannu Kadan Indigenous Association for obtaining a letter of consent, which was attached to the submission process to the CEP/UFRR. The samples were collected at the Emergency Service Cosme e Silva and Sayonara Health Unit from December 2021 to June 2022, due to the seasonal increase in the number of malaria cases in Roraima during this period. Both health centers are located in the west zone of the city and have the highest number of malaria reports according to the Malaria Epidemiological Surveillance Information System (Sivep-Malaria) (Figure 1).

Individuals over the age of 18 who had been diagnosed with *P. vivax* malaria or mixed malaria (*P. vivax* + *P. falciparum*), and had been diagnosed through thick blood smears, were included. The non-inclusion of individuals under 18 years old does not represent a limitation of this study, because the considerable increase in malaria cases occurred in mining areas where children under 18 years of age do not have access. Individuals who could not read or refused to sign the free and informed consent form (TCLE) were also excluded from this study. After signing the consent form, an epidemiologic questionnaire with questions about the person and malaria was completed.

Blood was collected by venipuncture of 5 mL of peripheral blood. A portion of the blood (approximately 50 microliters) was transferred directly from the syringe to filter paper (Whatman 903 Protein Saver Cards, Merck, Burlington, MA, USA) and the remainder to a Vacutainer tube (Becton, Dickinson & Company, Franklin Lakes, NJ, USA) containing EDTA.

All participants were treated according to the National Malaria Control Program (PNCM) protocol for non-severe malaria, which includes the administration of a combination of CQ for 3 days (10 mg/kg on day 1 and 7.5 mg/kg on days 2 and 3) and PQ for 7 days (0.5 mg/kg/day) [30].

According to the PNCM, cure control should be assessed through the cure verification slide (CVS) on days 3, 7, 14, 21, 28, 42, and 63 after the start of treatment, according to the operational capacity of the local health network. Collections on D3 and D28 should be prioritized for P. vivax infections. The day the diagnosis is made and treatment begins is considered day zero (D0) [31,32].

To identify the CVS of the participants infected by parasites with target mutations, a search was carried out in Sivep-Malaria for one year before and one year after the date of sample collection. To investigate malaria hospitalization cases after diagnosis, a search was carried out on Sivep-Malaria at the Notification Unit of the Roraima General Hospital, a state reference for severe malaria. The deaths were investigated by a search carried out on the Mortality Information System (SIM).

Relapse or recurrence was considered the reappearance of asexual parasitemia with or without symptoms after treatment due to the following: (i) recrudescence (incomplete clearance of asexual parasites after antimalarial treatment within 28 days); (ii) relapse (arising from hypnozoites between 28 and 60 days); or (iii) reinfection (after 60 days) [31,32].

The blood samples collected in vacutainer tubes and on filter paper were transported to the Molecular Biology Laboratory (LaBMol) of the Center for Biodiversity Studies (CBio) at the Federal University of Roraima (UFRR). The samples collected in the tubes were centrifuged at 3000× *g* for 10 min to remove the plasma and the cryopreservation solution glycerolyte 57 (Baxter, Minato City, Japan) was added to the “red blood cell concentrates” (containing leukocytes and platelets) volume by volume (*v*/*v*), followed by aliquoting each sample. The aliquots with the cryopreservation solution were stored at −20 °C in racks and packed in individually labeled plastic bags until the deoxyribonucleic acid (DNA) was extracted.

DNA extraction was carried out using the column technique (centrifugation method), using the QIAamp DNA Blood Mini Kit (Qiagen, Hilden, Germany), according to the manufacturer’s instructions, from a volume of 500 µL of the sample.

The methodology used for the PCR (polymerase chain reaction) of the *pvmdr1* gene was performed according to the protocol previously described, with the following primers: F: 5′-ATAGTCATGCCCCAGGATTG-3′ and R: 5′-ACCGTTGGTCTGGACAAGTAT-3′ [33].

A mixture of PCR reagents was prepared for a final volume of 50 µL, with 26.75 µL of ultrapure water, 6 µL of MgCl2 (25 mM), 5 µL of PCR Buffer II (10×), 5 µL of deoxynucleotide triphosphates (dNTPs) (8 mM), 1 µL of each primer (10 pmol), and 0.25 µL of AmpliTaq Gold DNA Polymerase (250 U). Finally, 5 µL of DNA was added to each mixture. The PCR conditions in the thermal cycler included the following: initiation at 95 °C for 10 min; 40 cycles with denaturation at 94 °C for 15 s; primer annealing at 60 °C for 30 s; extension at 72 °C for 1 min; and final elongation at 72 °C for 7 min. The final product was a 762 pb fragment amplified to analyze T958 **M**, Y976**F**, and F1076**L** in the *pvmdr1* gene.

The PCR reaction to amplify the *pvcrt-o* gene was based on the protocol previously described, with the following primers: F: 5′-AAGAGCCCGTCTAGCCAT CC-3′ and R: 5′-AGTTTCCCTCTACAC CCG-3′ [21]. To the reagent mixture with a final volume of 42 µL, 23.75 µL of ultrapure water, 4 µL of MgCl_2_ (25 mM), 4 µL of PCR Buffer II (10x), 2 µL of deoxynucleotide triphosphates (dNTPs) (8 nM), 2 µL of each of the primers (10 pmol), and 0.25 µL of AmpliTaq Gold DNA Polymerase (250 UI) were added. A total of 4 µL of DNA was added to each PCR reagent mixture. The PCR conditions in the thermal cycler included initial heating at 95 °C for 10 min, followed by 35 cycles with denaturation at 94 °C for 30 s, primer annealing at 61 °C for 1 min, extension at 72 °C for 1 min, and final elongation at 72 °C for 7 min. The final product of the PCR reaction (amplicon) generated a fragment of 1186 bp.

Three controls were used to ensure the accuracy and reliability of the results during the amplification of the target genes. The positive control consisted of a *Plasmodium vivax* sample diagnosed by molecular and microscopic examinations, whose target sequence amplification was confirmed by DNA sequencing. The negative control comprised a sample from a clinically healthy individual with no malaria history. Additionally, a blank control containing all test reagents but not the DNA, replaced by ultrapure water, was used.

To obtain the target sequences of the isolates, the PCR products were purified using the Wizard^®^ Kit, Promega, Madison, WI, USA, according to the manufacturer’s instructions. The sequencing reaction was performed using the Big DyeTM Terminator Cycle Sequencing Ready Reaction version 3.1 kit (Applied Biosystems, Waltham, MA, USA), and 3.2 pmol of forward and reverse primers were used separately in the reactions.

To investigate SNPs in the *pvmdr1* and *pvcrt-o* genes, the sense and antisense sequences of the samples and the reference sequence were aligned and analyzed with a ClustalW multiple sequence aligner in BioEdit software version 7.7.1 (North Carolina State University, Raleigh, NC, USA). The Salvador 1 (Sal-1) strain was used as the reference sequence (GenBank Accession No. AF314649.1 for *pvcrt-o* and No. AY571984.1 for *pvmdr1*).

The information from the questionnaires and the laboratory analysis results were entered and tabulated in the Excel program (Microsoft Office 365 v16.78.3^®^). The maps were drawn up using the QGIS program version 3.28.10. Mining areas in Roraima were obtained from Mapbiomas [34]. The geopolitical limits of Brazil and Indigenous lands were accessed on the IBGE website [35].

## 3. Results

Samples were collected from 164 participants, of whom 153 had *P. vivax* and 11 (7%) had mixed malaria (*P. vivax* + *P. falciparum*).

The participants ranged in age from 18 to 67, and 82% (135/164) were men. The most commonly reported symptoms were fever at 91.5% (150/164), headache at 86% (141/164), chills at 67% (110/164), abdominal pain at 55% (90/164), and nausea/vomiting at 43% (70/164). Detailed epidemiological information on these patients was recently published [36].

Regarding the main activity carried out in the 15 days before the onset of symptoms, 96% (157/164) of the participants reported gold mining, mainly in the municipalities of Alto Alegre with 76% (120/157) of the participants, and Mucajaí with 18.5% (29/157) of them. Agriculture accounted for 3% (5/164) of the participants, with 40% (2/5) in the municipality of Mucajaí and 20% (1/5) in the Alto Alegre, Cantá, and Caroebe municipalities. Hunting/fishing and tourism, with a percentage of 0.6% (1/164), occurred in participants from the municipalities of Alto Alegre and Mucajaí, respectively (Table 1).

The *pvcrt-o* gene was amplified in 94% (154/164) of the samples, and 99% (151/154) of the amplified products were sequenced. Of all the samples sequenced for the *pvcrt-o* gene, 87% (131/151) were identical to the Sal 1 strain, used as a wild-type reference for CQ-sensitive parasites. The lysine insertion (codon AAG) at position 10, called the K10 insertion, was identified in 13% (20/151) of the sequenced samples. The K10 insertion was detected in 25% (1/4) of the samples from participants engaged in agriculture, 100% (1/1) of those engaged in hunting/fishing, and 12% (18/145) of those engaged in illegal mining (Table 2). Despite the small number of samples, the K10 insertion was more frequent in prospectors (18) than in farmers (1) or hunters/fishermen (1).

The two participants who carried parasites with a K10 insertion in the *pvcrt-o* gene performed farming activities (1) and hunting/fishing activities (1) in the municipalities of Mucajaí and Alto Alegre, respectively, while the three farmer participants who carried parasites without a K10 insertion were probably infected in the municipalities of Cantá, Caroebe, and Mucajaí. The only participant who reported tourism carried parasites without a K10 insertion in the *pvcrt-o* gene, and the probable place of infection was the municipality of Mucajaí (Table 1 and Table 2).

Of all the participants who reported mining activities in the Yanomami Indigenous area of Roraima, the municipality of Alto Alegre ranked first, with 14% (15/109) of the parasites having a K10 insertion. The second-ranked municipality was Mucajaí, where 7% (2/28) of *P. vivax* samples showed K10 insertion. The only sample from a mining site in the municipality of Amajarí had no K10 insertion (Table 1 and Table 2).

Among the 110 individuals infected in Alto Alegre, 15% (16/110) had parasites carrying a K10 insertion. Of all the samples with probable infection in Mucajaí, 10% (3/31) had a K10 insertion. No K10 insertion was detected in the four samples from the municipalities of Amajarí, Cantá, and Caroebe. K10 was also not detected in samples of the four participants infected in Venezuela, but it was noted in 1/3 of the samples from participants infected in Guyana (Figure 2). In short, the K10 mutant was present in Alto Alegre, Mucajaí, and Guyana, and seems to be more fixed in Alto Alegre.

The *pvmdr1* gene was sequenced in 80 samples. Mutations in the entire fragment were analyzed, including the three codons (T958**M**, F1076**L**, and Y976**F**) potentially associated with the phenotype of RCQ in *P. vivax.* The T958**M** mutation (**M**YF haplotype)—in other words, threonine replaced by methionine at codon 958—was found in 92.5% (74/80) of the sequenced samples, while the wild-type TYF haplotype was identified in 7.5% (6/80) of the samples. The F1076**L** and Y976**F** mutations were absent in all the samples (Table 3).

When relating the sequenced samples of the *pvmdr1* gene and the main activity carried out by the patient in the 15 days before the onset of symptoms, the only participant who reported agriculture carried a parasite with the single mutation **M**YF haplotype (958M) from the municipality of Caroebe. The only participant who reported hunting/fishing carried a parasite with the wild TYF haplotype from the municipality of Alto Alegre. Concerning gold mining, 94% (73/78) of the participants carried parasites with the **M**YF haplotype from the municipalities of Alto Alegre (55), Mucajaí, and the neighboring country Venezuela (1), and 6% (5/78) with the wild TYF haplotype from Alto Alegre (3) and Mucajaí (2) (Table 3 and Figure 3).

The distribution of the *pvmdr1* gene haplotypes by probable site of infection reveals that the **M**YF haplotype is predominant in the municipalities of Alto Alegre (93%; 55/59) and Mucajaí (90%). The only sample from the municipality of Caroebe and neighboring Venezuela also had the **M**YF haplotype. The wild TYF haplotype was only present in a small number of samples from Mucajaí (11%) and Alto Alegre (5%) (Figure 3).

Regarding mining activities, the parasite haplotype was the **M**YF in the only sample from Venezuela. In parasites transmitted in mining Yanomami Indigenous areas of Alto Alegre, 95% (55/58) of the samples carried the **M**YF haplotype. Similarly, in infections from the Mucajaí, the MYF haplotype predominated (89%; 17/19) (Figure 3).

A total of 79 samples were sequenced for both the pvcrt-o and pvmdr1 genes. In addition to parasites with a single mutation in pvcrt-o or pvmdr1, five patients were infected with parasites carrying double mutants: insertion of K10 in the pvcrt-o gene and the **M**YF (958**M**) haplotype in the pvmdr1 gene. These patients reported mining activities in the 15 days preceding symptoms in the municipalities of Mucajaí (1) and Alto Alegre (4).

These data show that around 86% (68/79) of the parasite samples carried the **M**YF haplotype and lacked the K10 insertion. In contrast to the K10 insertion, the MYF mutant parasites are present in all studied municipalities (Figure 4).

No case notifications in the 28 days preceding or following the date of sample collection of the 18 patients carrying parasites with the pvcrt-o K10 insertion were registered in Sivep-Malaria, therefore showing no recrudescence episodes. However, probable cases of relapse in 11% (2/18) and reinfection in 39% (7/18) of those who reported mining activities were recorded (Table 4).

None of these participants were hospitalized or died during this study. The use of antimalarials as prophylaxis was reported by only one participant in mining in the municipality of Mucajaí (Table 4).

We also searched for registers in Sivep-Malaria of participants carrying parasites with the **M**YF haplotype of the pvmdr1 gene within 28 days before or after the date of sample collection, and no cases in the recrudescence period were observed in these patients. However, among them, the miner patients had 12% (9/73) cases of probable relapse and 29% (21/73) of reinfection. There were no records of hospitalization or death among these participants. Only 12% (9/73) of the participants with mining activities in the municipalities of Alto Alegre (7) and Mucajaí (2) reported use of antimalarials as prophylaxis.

## 4. Discussion

The emergence of CQR in *P. vivax* presents a significant challenge for eliminating malaria in Brazil by 2035. This species is responsible for the largest malaria burden in the legal Amazon. Moreover, the complex biology of *P. vivax* and the limited availability of laboratory research tools make it difficult to identify cases of antimalarial resistance in this parasite. Monitoring molecular markers to identify mutations related to antimalarial resistance over time can provide essential information to identify effective treatment policies and help determine the change of first-line drugs according to the local reality [22].

Detecting *P. vivax* CQR is complex due to the difficulty in distinguishing whether relapses or recurrences of the disease are due to relapse (related to hypnozoites), recrudescence (related to antimalarial resistance), or reinfection [7]. This difficulty is greater when monitoring occurrences in individuals conducting mining activities, as they often return to the place of infection. The illegal mining sites in Roraima are located in isolated areas of forest in the Yanomami Indigenous land, which comprises the municipalities of Amajarí, Alto Alegre, Mucajaí, Iracema, and Caracaraí. In these locations, the miners have no access to the Brazilian Health Unic System (SUS) healthcare diagnosis network [28]. When they travel to Boa Vista for malaria diagnosis and treatment, the gold miners return to the mines immediately after receiving diagnosis and antimalarial drugs, making it impossible to monitor the clinical efficacy of CQ through the negativity of parasitemia (CVS).

Thus, to identify the cure and whether there was any progression to the severe form of the disease, resulting in hospitalization or death, we consulted the SUS Health Information Systems (SIS), the Mortality Information System (SIM), and SIVEP-Malaria. This search proved to be an important strategy for investigation integrating research with local surveillance and minimizing SUS costs for monitoring the patients.

Few single-nucleotide polymorphisms (SNPs) have been reported in the *pvcrt* gene, unlike *pfcrt*, its ortholog in *P. falciparum*. The most common polymorphism in the *pvcrt* gene is a lysine insertion (codon AAG) at position 10 (K10), which was proposed to be associated to CQR [22,27,37].

In this study, we found a K10 insertion in the *pvcrt-o* gene in 13% (20/151) of the sequenced parasite samples, and no recrudescence, hospitalization, or death episodes in the patients infected with parasites carrying this mutation was noticed. Interestingly, the K10 insertion was only identified in parasites from locations with intense mining activity, such as Alto Alegre and Mucajaí, and in Roraima and the neighboring country of Guyana, suggesting that this mutant is becoming established in these areas because of an intense flow of individuals from different areas and possibly reflecting the greater diversity of *P. vivax* in areas with greater transmission.

In previous studies with patients from Acre, Amazonas, Amapá, Pará, Rondônia, and Roraima, the K10 insertion was not also associated with CQR [38]. The same was true in Manaus, where the K10 insertion was not related to in vitro *P. vivax* resistance to CQ [39], and in French Guiana, where no polymorphism in the *pvcrt-o* and *pvmdr1* genes was identified in CQR parasites [40].

In Southeast Asia parasites, the K10 insertion has been observed in prevalences ranging from 9.4% in India to 72% in Myanmar. However, no association between the presence of K10 insertion and in vitro *P. vivax* resistance to CQ was shown [27,41,42]. In fact, unlike what happens with *P. falciparum*, polymorphisms in *pvcrt* do not seem to be good molecular markers for monitoring CQR [38,40].

Regarding the *pvmdr1* gene, in patients carrying parasites with the T958M mutation (92%), no recrudescence or progression to severe malaria resulting in hospitalizations and deaths was identified through research into the search platforms. This finding is supported by other studies which have shown that the T958M mutation allele is a majority in parasite populations in endemic areas of Brazil, French Guiana, Asia, Pakistan, Afghanistan, Sri Lanka, Nepal, Sudan, São Tomé, and Ecuador, and that its presence is not associated with CQR [18,21,38,40,41,43,44,45,46].

The double mutation haplotype with amino acid changes in Y976F and F1076L, in particular, has already been cited as a possible marker of resistance to CQ [33,46]. However, this haplotype can also be found in regions with no reported cases of CQR, making its association with drug resistance uncertain and, consequently, its applicability as a marker of chemoresistance of *P. vivax* to CQ unprovable [38,46].

In the present study, no haplotypes with a double mutation profile (T958M + F1076L or Y976F + F1076L) were found in Roraima, Venezuela, or Guyana. Double mutants have not been previously identified in Roraima, although they have been found in patients from the state of Amazonas who responded well to treatment with CQ [38].

Parasites from French Guiana may have double T958M/F1076L and triple T958M/Y976F/F1076L mutated haplotypes in the *pvmdr1* gene [44]. However, the association between clinical response to the drug and/or in vitro susceptibility has never been demonstrated.

These data seem to suggest that the presence of these mutations in the *pvcrt-o* and *pvmdr1* genes is due to the remarkable genetic diversity of *P. vivax*, and that these polymorphisms have no implications for the phenotype of CQR parasites. Indeed, identifying these haplotypes in *P. vivax* parasite populations circulating in the gold mining individuals in regions representative of a scenario of high transmission could be an important database for analyzing these alleles over time.

Over two decades of research using molecular markers orthologous to *P. falciparum* into the resistance of *P. vivax* to antimalarials, the findings have shown no or a weak relationship with resistance to CQ, potentially leading to false conclusions that could impact national policy for the treatment of the disease [16,22].

There are marked differences in the topologies and number of SNPs in the *crt-o* and *mdr1* genes between *P. vivax* and *P. falciparum*, which reinforces the idea that other genes may be involved in the CQR phenotype in *P. vivax*. Therefore, understanding the molecular mechanisms of antimalarial resistance in *P. vivax* and investigating candidate genes to monitor CQR through ex vivo assays and sequencing could help identify genes other than these *P. falciparum* orthologs [22,38].

Despite the comparable selection pressure from the massive use of CQ, *P. vivax* CQR was only reported in 1989, whereas in *P. falciparum* it has been evident since the late 1950s. This can be explained by the differences in genetic determinants and molecular mechanisms of CQR in *P. falciparum* and *P. vivax* parasites [47,48] beyond the lower parasite biomass, the gametocytes production at the beginning of the infection, and the recurrence of hepatic hypnozoites. These conditions allow the parasite to be transmitted before the start of treatment or after the concentration of the drug has decreased. Furthermore, it has been suggested that the use of PQ may have the potential to reduce the transmission of CQR parasites [49].

Understanding the evolutionary and population dynamics of antimalarial resistance in *P. vivax* will be crucial for strengthening molecular surveillance, both to identify when these alleles arise and to understand how they move through and between populations [22,50]. This is probably especially true in the state of Roraima, where the dynamics of migratory flows and mining activities make malaria elimination an even more challenging goal.

Similar to Brazil, French Guiana is experiencing intense gold mining and human migration between countries in the Guiana Shield. This raises concerns about the spread of CQR *P. vivax* isolates, making surveillance and detection of resistant parasites critical [44].

Due to the remote mines’ geographical location, when miners are suspected to be infected with *Plasmodium*, mainly due to the presence of fevers and chills, they buy from the clandestine market the antimalarial Artecom^®^ (Dihydroartemisinin–Piperaquine/DHA-PQP) and take one dose/tablet of the drug, just to relieve symptoms for a fast return to the gold mines, and do not undergo complete treatment, highlighting the importance of molecular antimalarial resistance surveillance in these areas. This antimalarial, of known low quality, is unregistered in Brazil and illegally enters the country through the borders of Suriname, Guyana, and French Guiana [28,36,51].

WHO recommends DHA-PQP for the treatment of CQ-resistant *P. vivax* cases. However, no clinical trial has been reported to assess its efficacy in the Americas. An open randomized clinical trial of Eurartesim^®^ vs. CQ and Primaquine (PQ) for the radical cure of vivax malaria in Manaus, Brazil, was recently concluded [52]

In relation to *P. vivax* recurrence between 5 and 60 days, in Brazil, the treatment guidelines recommend artemether/lumefantrine or artesunate/mefloquine for 3 days to clear asexual parasites and PQ for 14 days to eliminate hypnozoites and prevent relapse, a treatment termed radical cure [30]. However, nonadherence to the primaquine regimen harms the effectiveness of the treatment. Tafenoquine (TFQ) is a longer-acting 8-aminoquinoline. It has about a 15-day half-life with a single-dose treatment. Thus, it can replace PQ for facilitating patient adherence and radical cure, avoiding the chances of *P. vivax* relapse and serving as a new ally in the search for the elimination of malaria [53].

In Brazil, TFQ was incorporated into the SUS in 2023, along with G6PD deficiency tests to be prescribed only to people over 16 years old and with at least 70% G6PD activity [54]. In Roraima, in April 2024, the Special Indigenous Health District Yanomami (DSEI-Yanomami) started implementing TFQ due to the greater difficulty in adhering to long-term PQ treatment among Indigenous people.

A possible limitation that could lead to incorrect conclusions is related to the treatment protocol because all the patients received not only CQ but also a PQ radical curative regimen, which would suggest that PQ would have eliminated resistant CQ parasites and, therefore, the presence of CQR parasitic populations could not be ruled out. However, CQ, with or without PQ, produces a rapid parasite clearance in sensitive parasites. In contrast, drugs with later-stage specificity, such as PQ, give slow parasite clearance rates [55]. Although there is some in vitro evidence of synergy between PQ and CQ against *P. falciparum* schizonts [56], there is no evidence of synergy between these two drugs against *P. vivax* asexual blood stages [16].

## 5. Conclusions

The results of this study reinforce the continuous use of CQ in Roraima and also corroborate that mutations in the *pvcrt-o* and *pvmdr-1* genes have no predictive potential of the CQR *P. vivax* phenotype in Brazilian endemic areas. The molecular mechanisms of antimalarial resistance in *P. vivax* and the difference in the evolutionary dynamics of the *P. vivax* and *P. falciparum* populations suggest that molecular markers associated with *P. vivax* chemoresistance to CQ may lie beyond the *P. falciparum* orthologues.

## Figures and Tables

**Figure 1 microorganisms-12-01680-f001:**
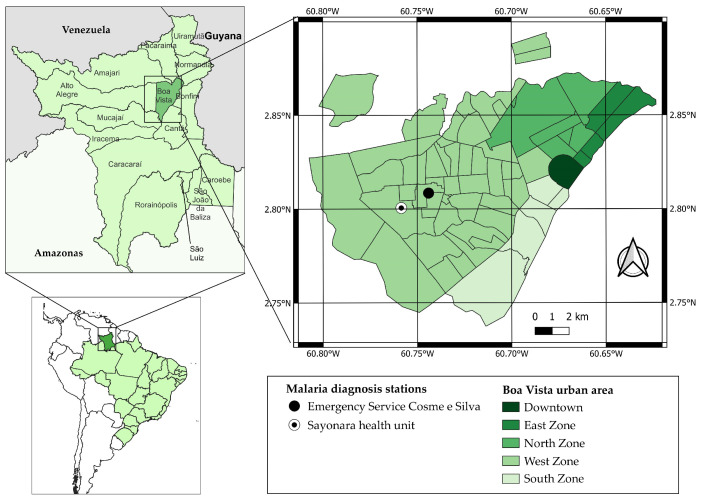
Location map of Roraima, Boa Vista, the urban area of Boa Vista, and sample collection sites.

**Figure 2 microorganisms-12-01680-f002:**
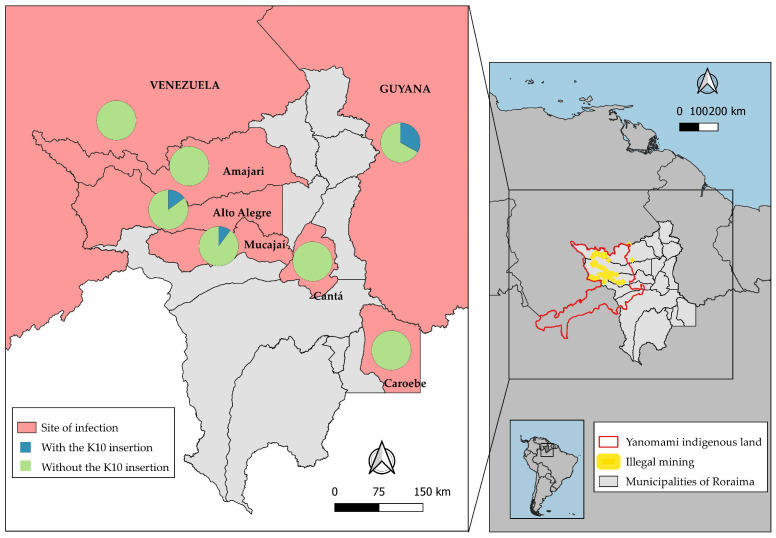
Distribution of K10 insertions (codon AAG) in the *pvcrt-o* gene in *P. vivax* samples, according to probable site of infection (n = 151).

**Figure 3 microorganisms-12-01680-f003:**
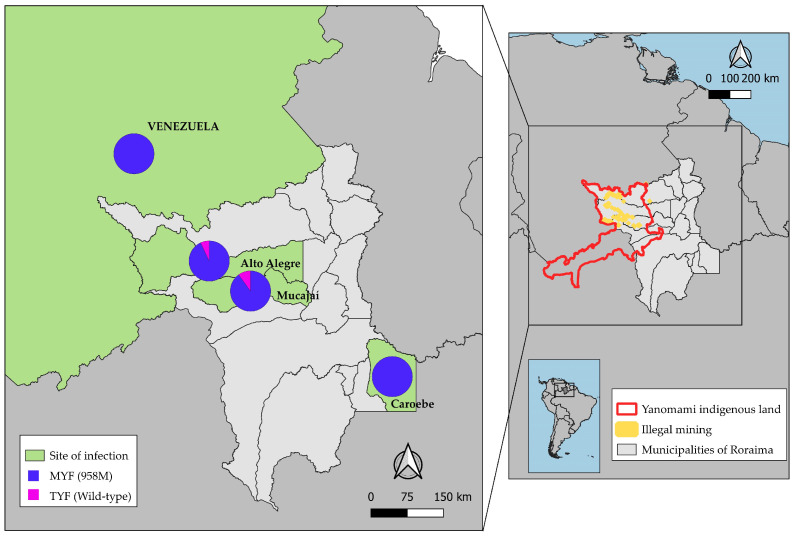
Distribution of *pvmdr1* gene haplotypes in *P. vivax* samples, according to probable site of infection (n = 80).

**Figure 4 microorganisms-12-01680-f004:**
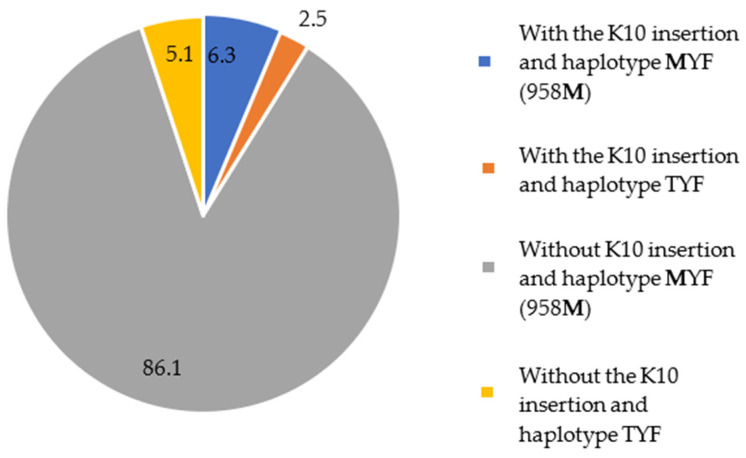
Percentage of *pvcrt-o* and *pvmdr1* gene alleles in the 79 samples sequenced for both genes.

**Table 1 microorganisms-12-01680-t001:** Distribution of study participants according to the main activities performed in the 15 days before symptoms and the probable site of infection (n = 164).

Probable Site of Infection	Illegal Mining		Agriculture		Hunting/Fishing		Tourism		Total	
	N	%	N	%	N	%	N	%	N	%
Alto Alegre	120	76.4	1	20	1	100	0	0	122	74.4
Amajari	1	0.6	0	0	0	0	0	0	1	0.6
Cantá	0	0.0	1	20	0	0	0	0	1	0.6
Caroebe	0	0.0	1	20	0	0	0	0	1	0.6
Mucajaí	29	18.5	2	40	0	0	1	100	32	19.5
Guyana	3	2	0	0	0	0	0	0	3	1.8
Venezuela	4	2.5	0	0	0	0	0	0	4	2.5
Total	157	100	5	100	1	100	1	100	164	100

**Table 2 microorganisms-12-01680-t002:** Distribution of K10 insertions in the *pvcrt-o* gene (n = 151), according to the main activity carried out by patients in the 15 days before symptoms occurred.

K10	Agriculture		Illegal Mining		Hunting/ Fishing		Tourism		Total	
	N	%	N	%	N	%	N	%	N	%
With	1	25	18	12.4	1	100	0	0	20	13.2
Without	3	75	127	87.6	0	0	1	100	131	86.8
Total	4	100	145	100	1	100	1	100	151	100

**Table 3 microorganisms-12-01680-t003:** Haplotype distribution in the *pvmdr1* gene (n = 80), according to the main activity carried out by the study participants in the 15 days before symptoms occurred.

Haplotypes	Agriculture		Illegal Mining		Hunting/ Fishing		Total	
	N	%	N	%	N	%	N	%
**M**YF (**958M**)	1	100	73	94	0	0	74	92.5
TYF	0	0	5	6	1	100	6	7.5
Total	1	100	78	100	1	100	80	100

**Table 4 microorganisms-12-01680-t004:** Distribution of parasites with a K10 insertion in the *pvcrt-o* gene, according to activities in the last 15 days, and notifications in Sivep-Malaria about hospitalization, death, and use of antimalarials for prophylaxis (n = 20).

Variables	Agriculture (n = 1)		Illegal Mining (n = 18)		Hunting/Fishing (n = 1)		Total (n = 20)	
	N	%	N	%	N	%	N	%
Sample Collection	1	100	9	50	1	100	11	55
Recrudescence (<28 days)	0	0	0	0	0	0	0	0
Probable Relapse (29 to 60 days)	0	0	2	11	0	0	2	10
Probable Reinfection (>60 days)	0	0	7	39	0	0	7	35
Hospitalization	0	0	0	0	0	0	0	0
Death	0	0	0	0	0	0	0	0
Use Antimalarials Prophylaxis	0	0	1	5.6	0	0	1	5

## Data Availability

Sequences were deposited in Genbank^TM^ with *pvmdr1* accession number PP693801-PP693880 and *pvcrt-o* accession number PP681704-PP681857.

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
