# Peer review of "Investigation of Mutations in the crt-o and mdr1 Genes of Plasmodium vivax for the Molecular Surveillance of Chloroquine Resistance in Parasites from Gold Mining Areas in Roraima, Brazil"

_microorganisms, 2024, doi:10.3390/microorganisms12081680_

Round 1

Reviewer 1 Report

Comments and Suggestions for Authors

This is an interesting and insightful data on the molecular marker of P. vivax mediated malaria in select Brazilian communities. The manuscript as such has sufficient data to to support conclusions. I have few comments to make. 

The CQR resistance in plasmodial species is a well documented one. The WHO suggested treatment for CQR malaria is a combination therapy containing piperaquine as one of the active agents. Is there any data available for the present study on the resistance of piperaquine, along with primaquine?

Is there reports on artemisinin and artemether treatment results to CQR in the geographical region reported here?

A recent initiative from GSK and medicine for malaria venture has introduced tafenoquine as a single dose cure for P. vivax malaria relapse and resistance malaria. This manuscript needs to address these latest developments in light of their own data.  

Comments on the Quality of English Language

The English language is fine, but please edit the abstract section by removing the numbers. An abstract is a continuous summary of your manuscript not bullet points. 

Author Response

Point-by-point response to Comments and Suggestions for Authors

Comments 1: The WHO suggested treatment for CQR malaria is a combination therapy containing 
piperaquine as one of the active agents. Is there any data available for the present study on the resistance of piperaquine, along with primaquine?
Response 1: Thank you for pointing this out. We agree with this comment. No data is available 
concerning the resistance of piperaquine along with primaquine in Brazil. A sentence on this sense 
was introduced on page 11, in lines 427 to 439.

Comments 2: Is there reports on artemisinin and artemether treatment results to CQR in the 
geographical region reported here?
Response 2: No, there isn’t. A sentence about the Brazilian guideline treatment for P. vivax 
recurrence was introduced on page 11, in lines 436-439.

Comments 3: A recent initiative from GSK and the Medicine for Malaria Venture has introduced 
tafenoquine as a single-dose cure for P. vivax malaria relapse and resistance. This manuscript needs to 
address these latest developments in light of their own data. 
Response 3: A sentence on this sense was added at the MS on page 11, in lines 439-448.

Response to Comments on the Quality of English Language
Point 1: The English language is fine, but please edit the abstract section by removing the numbers. 
An abstract is a continuous summary of your manuscript not bullet points.
Response 1: The numbers were removed from the abstract

Reviewer 2 Report

Comments and Suggestions for Authors

The manuscript microorganisms-3156385 claims that the mutations in the pvcrt-o and pvmdr-1 genes have no potential to predict chloroquine resistance in P. vivax. The manuscript is scientifically significant and has a direct impact on public health. The analysis is clear, and the data are well-discussed. Thus, I recommend the publication after minor revisions as follows:

1) Section 4 provides the limitations and future perspective of this work.

2) In the discussion it is not clear the impact that the obtained results might have on the public health of Roraima. Please, better explore it.

3) Please, provide the controls used in the assays.

4) The references must be in accordance with the journal’s rules. Please, check it.

Author Response

Point-by-point response to Comments and Suggestions for Authors
Comments 1: Section 4 provides the limitations and future perspective of this work.
Response 1: Thank you for pointing this out. We agree with this comment. A sentence on this 
sense was added to the MS on page 11, lines 449-457. 

Comments 2: In the discussion it is not clear the impact that the obtained results might have on the 
public health of Roraima. Please, better explore it.
Response 2: A sentence on this sense was added at the MS on page 10, in the lines 391-393 and 
460.

Comments 3: Please, provide the controls used in the assays.
Response 3: A sentence on this sense was added to the MS on page 4, in lines 180-186.

Comments 3: The references must be in accordance with the journal’s rules. Please, check it.
Response 3: Agree. We have revised, and now, the references are following the journal’s rules
